# Relationship between Oxidative Stress and Left Ventricle Markers in Patients with Chronic Heart Failure

**DOI:** 10.3390/cells12050803

**Published:** 2023-03-04

**Authors:** Aušra Mongirdienė, Agnė Liuizė, Dovilė Karčiauskaitė, Eglė Mazgelytė, Arūnas Liekis, Ilona Sadauskienė

**Affiliations:** 1Department of Biochemistry, Medicine Academy, Lithuanian University of Health Sciences, Eiveniu Str. 4, LT-50103 Kaunas, Lithuania; 2Cardiology Clinic, University Hospital, Lithuanian University of Health Sciences, Eiveniu Str. 2, LT-50161 Kaunas, Lithuania,; 3Department of Physiology, Biochemistry, Microbiology and Laboratory Medicine, Institute of Biomedical Sciences, Faculty of Medicine, Vilnius University, M. K. Čiurlionio st. 21, LT-03101 Vilnius, Lithuania; 4Neuroscience Institute, Lithuanian University of Health Sciences Eiveniu Str. 4, LT-50103 Kaunas, Lithuania

**Keywords:** oxidative stress, left ventricle, protein carbonyls, nitrotyrosine, malondialdehyde, heart failure, oxHDL, total plasma antioxidant capacity

## Abstract

Oxidative stress is proposed in the literature as an important player in the development of CHF and correlates with left ventricle (LV) dysfunction and hypertrophy in the failing heart. In this study, we aimed to verify if the serum oxidative stress markers differ in chronic heart failure (CHF) patients’ groups depending on the LV geometry and function. Patients were stratified into two groups according to left ventricular ejection fraction (LVEF) values: HFrEF (<40% (*n* = 27)) and HFpEF (≥40% (*n* = 33)). Additionally, patients were stratified into four groups according to LV geometry: NG–normal left ventricle geometry (*n* = 7), CR–concentric remodeling (*n* = 14), cLVH–concentric LV hypertrophy (*n* = 16), and eLVF–eccentric LV hypertrophy (*n* = 23). We measured protein (protein carbonyl (PC), nitrotyrosine (NT-Tyr), dityrosine), lipid (malondialdehyde (MDA), oxidizes (HDL) oxidation and antioxidant (catalase activity, total plasma antioxidant capacity (TAC) markers in serum. Transthoracic echocardiogram analysis and lipidogram were also performed. We found that oxidative (NT-Tyr, dityrosine, PC, MDA, oxHDL) and antioxidative (TAC, catalase) stress marker levels did not differ between the groups according to LVEF or LV geometry. NT-Tyr correlated with PC (*r_s_* = 0.482, *p* = 0.000098), and oxHDL (*r_s_* = 0.278, *p* = 0.0314). MDA correlated with total (*r_s_* = 0.337, *p* = 0.008), LDL (*r_s_* = 0.295, *p* = 0.022) and non-HDL (*r_s_* = 0.301, *p* = 0.019) cholesterol. NT-Tyr negatively correlated with HDL cholesterol (*r_s_* = -0.285, *p* = 0.027). LV parameters did not correlate with oxidative/antioxidative stress markers. Significant negative correlations were found between the end-diastolic volume of the LV and the end-systolic volume of the LV and HDL-cholesterol (*r_s_* = –0.935, *p* < 0.0001; *r_s_* = –0.906, *p* < 0.0001, respectively). Significant positive correlations between both the thickness of the interventricular septum and the thickness of the LV wall and the levels of triacylglycerol in serum (*r_s_* = 0.346, *p* = 0.007; *r_s_* = 0.329, *p* = 0.010, respectively) were found. In conclusions, we did not find a difference in serum concentrations of both oxidant (NT-Tyr, PC, MDA) and antioxidant (TAC and catalase) concentrations in CHF patients’ groups according to LV function and geometry was found. The geometry of the LV could be related to lipid metabolism in CHF patients, and no correlation between oxidative/antioxidant and LV markers in CHF patients was found.

## 1. Introduction

Depending on its pathogenesis, heart failure is classified as either HF with preserved left ventricle ejection fraction (HFpEF) or HF with reduced ejection fraction (HFrEF), with HFpEF accounting for almost half of all cases of HF [1]. The exact sequence of events that contribute to the development and progression of chronic HF (CHF) remains to be elucidated. HFrEF is known to be the result of myocardial ischemia and infarction, while HFpEF is associated with older age, dysregulated metabolism, and chronic hypertension, which contribute to oxidative stress and myocardial remodelling and dysfunction [2]. Oxidative stress (an imbalance between the increased formation of reactive oxygen species (ROS) and the elimination or neutralization of ROS by an antioxidant system [3]) is proposed as an important player in the development of CHF [4]. It correlates with left ventricle (LV) dysfunction and hypertrophy in the failing heart [5] and is involved in ventricular remodelling [6].

In a systemic review, Martins. et al. concluded that antioxidants have the potential to become of a therapeutic strategy against this important pathological condition [7,8]. However, evidence for the benefit of antioxidant therapies in clinical trials is mainly from animal models and is sparse [7,9]. Furthermore, the reason for the discrepancy in antioxidative stress therapies was raised because it could be that only specific patient groups benefit from antioxidative stress treatments [10]. Therefore, more studies are needed to better understand the role of oxidative stress as a therapeutic target for cardiac remodelling.

There are data suggesting that in the case of HFrEF, myocardial remodelling is stimulated by ROS formed in heart cells, while in the case of HFpEF, it is stimulated by ROS that have increased in the myocardium due to external changes [9]. Metabolic syndrome, chronic hypertension, and other reasons are implied to dysregulate the human antioxidant system [8] and result in increased and oxidised proteins and lipids in the blood. ROS is revealed to directly activate GTP-binding proteins in myocytes and promote both hypertrophic growth signalling and apoptosis [11,12].

Nitrotyrosine (NT-Tyr) and dityrosine are products of tyrosine oxidation [13]. NT-Tyr is presented as a potential marker of oxidative stress associated with tissue damage and has a potential role as an inflammatory mediator in coronary artery disease (CAD) [14]. Protein carbonylation is the most well-known type of protein oxidation resulting in irreversible loss of protein function [15]. Catalase is an enzyme of the antioxidant system [16,17]. Malondialdehyde (MDA) is presented as an important indicator of lipid peroxidation [18]. The total antioxidant capacity (TAC) reflects an estimation of the ability of different antioxidants and shows the net results of the complex interaction between oxidants and antioxidants [19].

Concentrations of the carbonyl oxidative stress markers protein (PC) and MDA in serum were shown to be higher in left ventricle (LV) hypertrophy of haemodialysis patients compared to a healthy person and were correlated with the geometry of the LV [20]. Therefore, we aimed to verify two hypotheses: 1) in HFpEF there should be more oxidised proteins and lipids in the blood than in HFrEF, 2) oxidative stress markers in serum could differ in patients depending on the LV geometry. In this case, the reduction in oxidative stress for patients of the specific group would be more appropriate. If the hypothesis is confirmed, it would be possible to study the possibility of reducing the effect of oxidants in the specific CHF group according to left ventricle ejection fraction (LVEF) or in the group according to the geometry of the LV to inhibit harmful remodelling and failure progression in specific patients with CHF.

## 2. Methods

### 2.1. Study Population

A total of 60 patients diagnosed with CHF, admitted to the Department of Cardiology at Kaunas Clinical Hospital of Lithuanian University of Health Sciences between 1 January 2016 and 1 March 2018, were included in the study. All patients gave their written consent. Inclusion criteria included no changes in functional class according to the New York Heart Association (NYHA) or medical treatment in the past 3 to 4 weeks and no new HF symptoms. The diagnosis of CHF was made following the guidelines for the diagnosis and treatment of heart failure approved by the European Society of Cardiology [21]. Patients with kidney failure (glomerular filtration rate (GFG) < 60 mL/min.), acute or chronic infection, acute coronary syndromes, diabetes mellitus, connective tissue disease, or smoking were excluded from the study.

The study group consisted of 26 (43.3%) women and 34 (56.7%) men whose age median (IQR) was 67.5 (22.5) years. First of all, patients were stratified into two groups according to left ventricular ejection fraction (LVEF) values: HFrEF (<40% (*n* = 27)) and HFpEF (≥40% (*n* = 33)). There was an equal distribution of patients in the NYHA classification categories (Class II-IV). Comparison of sociodemographic and clinical characteristics between the groups based on left ventricular ejection fraction values (reduced vs. midrange and preserved LVEF) showed a significantly higher number of men in a group of reduced (<40%) LVEF. Furthermore, in a group of mid-range and preserved LVEF, most subjects were assigned to NYHA Class II or Class III, while in a group of reduced LVEF, the majority of patients were assigned to NYHA Class IV, and these differences were statistically significant. The sociodemographic and clinical characteristics of the study group are presented in Table 1.

### 2.2. Tests and Blood Sampling

Transthoracic echocardiogram analysis and complete blood count test were performed after admission of the patients to the hospital. Blood samples were drawn from the forearm vein with a 20 G needle into 4.5 mL vacuum tubes with ethylendiamintetraacetic acid (EDTA) and without additives. Complete blood count testing was performed on a COULTER LH 780 hematological analyzer (Brea, CA, USA). Blood serum samples for oxidative (nitrotyrosine, dityrosine, protein carbonyl, malondialdehyde, oxidised HDL) and antioxidative (total plasma antioxidant capacity, catalase (CAT)) stress biomarkers were frozen at −80 °C until analysis.

Oxidative and antioxidative markers were measured in serum using commercial reagent kits: Human total antioxidant capacity ELISA Kit abx053643 (abbexa, United Kingdom), Carbonyl Protein ELISA K7870 (immune diagnostic AG, Stubenwald-Allee 8a, 64625 Bensheim, Germany), Human oxidised high-density lipoprotein (Ox-HDL0 ELISA Kit CSB-E16552h (Cusabio biotech Co, Wuhan, China) and Nitrotyrosin ELISA K7829 (immune diagnostic AG). CAT activity in serum was evaluated according to the method described in [22]. CAT activity was measured by hydrogen peroxide reaction with ammonium molybdate, which produces a complex that absorbs at a wavelength of light of 410 nm. The results were expressed in U/mg protein. Under these conditions, one unit of catalase (U) decomposes 1 mol of hydrogen peroxide per 1 min. The protein concentration in serum was measured using the Lowry method [23]. Samples for MDA were prepared and analysed according to the methodology of Khoschosorur et al. [24], using the HPLC method with fluorescence detection. Chromatographic separation was performed on the HPLC system (Shimadzu Nexera X2, Kyoto, Japan). A 20-μL sample was injected onto the HPLC column (Agilent Poroshell, Santa Clara, California, 120 EC–C18, 3 × 100 mm, 2.7 μm). The chromatographic isocratic separation was carried out with a binary mobile phase of methanol and 50 mM phosphate buffer, pH 6.8 (2:3, *v/v*). Fluorescence detection was performed at 230 nm excitation and 430 nm emission wavelengths. The average retention time of the malondialdehyde-thiobarbituric acid adduct was 1.63 min.

All investigations were approved and conducted in accordance with the guidelines of the local Bioethics Committee and adhered to the principles of the Declaration of Helsinki and Title 45, US Code of Federal Regulations, Part 46, Protection of Human Subjects (revised 15 January 2009, effective 14 July 2009). The study was approved by the Regional Bioethics Committee of the Lithuanian University of Health Sciences (No. BE-2-102, 20 December 2018).

### 2.3. Statistical Analysis

Statistical analysis was performed using the R software, version 4.2.2, R Core Team, 2021. The Shapiro–Wilk test was used to test the normality of the variables. Quantitative variables are presented as mean ± standard deviation (SD) for normally distributed variables, or median (interquartile range) (IQR) for non-normally distributed variables. For comparison of the median (IQR) or average ± SD values between the two groups, the Mann–Whitney U test or Student’s *t*-test were used. Comparison of median (IQR) values among three or more groups was performed using the Kruskal–Wallis test. Dunn’s post hoc test with Bonferroni correction was used for multiple comparisons. Pearson’s Chi-square test and post hoc analysis or Fisher’s exact test (when any expected frequency was less than or equal to 5) were used to compare categorical variables between the groups based on values of left ventricular ejection fraction or left ventricular geometry. Spearman’s rank coefficient (for non-normally distributed variables) or Pearson’s correlation analysis (for normally distributed variables) was used to quantify the strength of the correlation between continuous variables. The level of statistical significance was set at 0.05 for the two-tailed test.

## 3. Results

### 3.1. Characteristics of Study Groups according to LVEF

Data on drug use showed that more than half (56.7%) of patients receive angiotensin converting enzyme (ACE) inhibitors, 48.3% used β-blockers and 30% diuretics. Comparison of medication usage between groups based on left ventricular ejection fraction values revealed that there was no statistically significant difference in prescribed medications (Table 2).

### 3.2. Oxidative/Antioxidant Stress Markers in Groups according to LVEF

Oxidative (nitrotyrosine, dityrosine, protein carbonyl, malondialdehyde, oxidized HDL) and antioxidative (total plasma antioxidant capacity, catalase) stress biomarker levels were not significantly different in the two groups of different LVEF values (Table 3).

Additionally, we divided the entire study sample into two groups based on the median values of malondialdehyde, protein carbonyl, and oxidised HDL levels. No statistically significant differences were found in serum oxidative/antioxidant stress markers and values of LVEF between groups with different concentrations of malondialdehyde concentration (≤114.29 µg/L vs. >114.29 µg/L) (Table 4).

Comparison of oxidative/antioxidant stress markers and values of the LVEF between groups of different levels of protein carbonyl and oxidised HDL showed that the median concentration of NT-Tyr in serum was significantly higher in a group of patients with increased levels of PC (3.16 (2.09) nM vs. 4.46 (2.12) nM, *p* = 0.008) and oxHDL (3.42 (1.38) pg/L vs. 4.51 (2.45) pg/L, *p* = 0.004) levels. No significant differences in the levels of other oxidative stress parameters were found among participants with different PC and oxHDL levels (Table 5 and Table 6).

### 3.3. Characteristics of Study Groups according to LV Geometry

Comparison of sociodemographic and clinical parameters among the left ventricular geometry groups showed statistically significant differences in the proportion of men and women in the different left ventricular geometry groups. However, post hoc pairwise comparisons for the Chi-squared test revealed that these differences were not significant. The Kruskal–Wallis analysis indicated statistically significant differences in age and HDL cholesterol levels between the groups. Dunn’s post hoc test with Bonferroni correction for multiple comparisons showed significant differences in the age of subjects between the NG and cLVH groups, as well as between the eLVH and cLVH groups. Post hoc analysis of HDL cholesterol levels between left ventricular geometry groups yielded nonsignificant results (Table 7). Furthermore, the differences in medication usage among the different left ventricular geometry groups were not significant (Table 8).

### 3.4. Oxidative/Antioxidative Stress Markers in Groups according to the Geometry of the Left Ventricle

Additionally, we analysed the levels of oxidative/antioxidative stress markers in the groups based on the geometry of the left ventricle. However, no statistically significant differences were found between the groups in both oxidative parameters (nitrotyrosine, dityrosine, protein carbonyl, malondialdehyde, oxidised HDL) and antioxidative parameters (total plasma antioxidant capacity, catalase) (Table 9).

### 3.5. Correlation Analysis

The correlation analysis revealed a statistically significant negative association between PC concentration and LVEF (*r_s_* = 0.257, *p* = 0.047). However, since a significant difference was found in the distribution of men and women between the groups of reduced and mid-range or preserved LVEF, we calculated the partial Spearman correlation adjusted for gender. As a result, the previously observed association between LVEF and PC concentration fell to a nonsignificant level (*r_s_* = 0.240, *p* = 0.067) (Table 10). In the current study, no significant correlation was found between oxidative/antioxidative stress biomarkers and subjects’ age, systolic and diastolic blood pressure.

Analysis of the relationship between the levels of biomarkers of oxidative/antioxidative stress revealed a moderate and statistically significant association between the concentration of NT-Tyr and PC levels (*r_s_* = 0.482, *p* = 0.000098), as well as statistically significant correlation between NT-Tyr and oxHDL levels serum (*r_s_* = 0.278, *p* = 0.0314) (Table 11).

Spearman’s correlation analysis revealed weak but statistically significant positive associations between the age of the subjects and the HDL cholesterol level (*r_s_* = 0.315, *p* = 0.014) and between systolic blood pressure and total cholesterol concentration levels (*r_s_* = 0.255, *p* = 0.049). Investigation of the relationship between oxidative stress measures and lipid metabolism biomarkers showed significant positive correlations between MDA concentration and total cholesterol levels (*r_s_* = 0.337, *p* = 0.008), LDL cholesterol levels (*r_s_* = 0.295, *p* = 0.022), and non-HDL cholesterol levels (*r_s_* = 0.301, *p* = 0.019), as well as a negative correlation between nitrotyrosine and HDL cholesterol concentrations (*r_s_* = 0.285, *p* = 0.027) in the study group (Table 12).

The results showed that none of the echocardiographic characteristics were significantly correlated with oxidative/antioxidative stress markers (Table 13). However, most of the echocardiographic parameters, such as left ventricular end-diastolic dimension (LVEDD), interventricular septum thickness (IVST), posterior wall thickness (PWT), left ventricle wall thickness (LVWT), and relative wall thickness (RWT), correlated significantly with the age of the participants (*r_s_* = –0.335, *p* = 0.009; *r_s_* = 0.448, *p* = 0.0003; *r_s_* = 0.383, *p* = 0.003; *r_s_* = 0.434, *p* = 0.0005; *r_s_* = 0.461, *p* = 0.0002, respectively). Additionally, a significant negative association was observed between the end-diastolic dimension of the left ventricle and HDL cholesterol (*r_s_* = −0.256, *p* = 0.048). However, after adjustment for age, the correlation was found to be non-significant (*r_s_* = −0.168, *p* = 0.202). Furthermore, very strong statistically significant negative correlations were found between the end-diastolic volume of the left ventricular or the end-systolic volume of the left ventricular and HDL-cholesterol in the study group (*r_s_* = –0.935, *p* < 0.0001; *r_s_* = –0.906, *p* < 0.0001, respectively). Furthermore, the results showed significant positive associations between the thickness of the interventricular septum or the thickness of the left ventricle wall and the levels of triacylglycerol in blood serum (*r_s_* = 0.346, *p* = 0.007; *r_s_* = 0.329, *p* = 0.010, respectively) (Table 14).

## 4. Discussion

### 4.1. The Difference in Oxidative Stress/Antioxidant Markers between the CHF Groups

Regardless of our theoretical assumptions, there was no difference in oxidative/antioxidant markers between the groups according to the LVEF and the LV geometry. In the group of patients with higher levels of PC, the median concentration of NT-Tyr was significantly higher (4.46 (2.12) nM vs. 3.16 (2.09) nM, *p* = 0.008), as well as oxidised HDL (4.51 (2.45) pg/L vs. 3.42 (1.38) pg/L, *p* = 0.004). A moderate and statistically significant correlation was found between NT-Tyr concentration and PC levels (*r_s_* = 0.482, *p* = 0.000098). Therefore, our work supplements the knowledge that oxidants in the blood oxidise both proteins (as shown by markers of NT-Tyr and PC) and lipids (as shown by markers of MDA and HDL).

Research of oxidative stress/antioxidative status in CVD is mainly focused on markers of oxidation of proteins and lipids, antioxidant enzymes and antioxidant capacity.

To date, there are little data on serum NT-Tyr levels in CVD. Shishehbor et al. [14] revealed significantly increased levels of NT-Tyr in patients with CAD (*n* = 100, median 9.1 μmol/mol, *p* < 0.001) compared to healthy controls (*n* = 108, median 5.2 μmol/mol, *p* < 0.001). Ferlazzo et al. did not find a difference in plasma NT-Tyr levels between CHF patients and healthy people [25]. In our previous work, we found that dityrosine concentration was significantly higher in CHF patients compared to healthy individuals (*n* = 67, average 1.54 (0.48) relative units of fluorescence vs. *n* = 31, average 1.27 (0.53) relative units of fluorescence, *p* < 0,05) and increased with increasing serum hypochlorous acid concentration [26]. Grzegorz et al. showed that plasma NT-Tyr concentration was higher in 60 year old morbidly obese people than in obese 20–39 year old individuals [27]. Despite the difference in age between the groups according to the LV geometry, we did not find any differences in the NT-Tyr concentration between our CHF groups.

The concentration of MDA (a lipid oxidation marker) and PC (a protein oxidation mas) in serum has been shown to be higher, and the serum TAC concentration lower, in patients on hemodialysis with LV hypertrophy (*n* = 92) compared to normal LV geometry (NG, *n* = 12) [20]. MDA was found to be higher in cLVH (*n* = 33) compared to eLVH (*n* = 45), and PC was found to be higher in eLVH compared to cLVH. TAC was significantly lower in the eLVH, cLVH, and CR (*n* = 14; averages 2.23 (0.42), 2.38 (0.28), and 2.38 (0.2)) group compared to NG patients (2.90 (0.32); *p* < 0.001) [20]. According to these data, the authors suggest that oxidative damage of proteins is more important for the pathogenesis of eLVH, while oxidative damage of lipids could be more important to cLVH. However, our findings in the CHF groups do not agree with the findings of Zorica et al. in hemodialysis patients with LV hypertrophy.

Radovanovic et al. investigated lipid (8-epi-prostaglandin F2α, 8-epi-PGF2α, and MDA) and protein (protein thiol group group (P-SH)) oxidative stress markers in patients with different severities of ischaemic heart failure based on NYHA class and a control group. The authors found that MDA was significantly higher in NYHA class III and IV [28]. The levels of 8-epi-PGF2α concentration in 24 h urine samples were found to be significantly higher in patients with NYHA classes III and IV than in healthy subjects and patients with NYHA I and II. Significant differences in 8-epi-PGF2α excretion were also found between NYHA classes III and IV (*p* < 0.01), with the rise more pronounced in NYHA class IV patients. The P-SH content was significantly lower in patients with NYHA III and IV classes compared to controls [28]. Therefore, the results of MDA, 8-epi-PGF2α and P-SH could show the importance of lipid and protein oxidation in the worsening of LV function. In another study, Radovanovic et al. found a significant increase in plasma reactive carbonyl derivatives (RCD), indicating protein oxidation, in all groups of CHF patients compared to the control group [29]. P-SH levels were significantly lower in NYHA IV patients (*n* = 10) compared to controls (*n* = 69) and NYHA I/II (*n* = 11/71) in this study. As the functional class increased, the MDA increased steadily. The enzyme of the antioxidant system, glutathione peroxidase (GPX) activity in plasma from patients with severe CHF (NYHA III/IV class) was significantly lower compared to NYHA I/II, as well as in healthy individuals. Another enzyme of the antioxidant system, superoxide dismutase (SOD), was shown to increase in all groups of patients with CHF compared to healthy controls [29]. The authors concluded that their results showed a relationship between plasma markers of oxidative damage and stage-dependent progression of CHF and that the presence of oxidative products could define oxidative damage in the myocardium undergoing the remodelling process. Carbonyl stress, they stated, could be implicated in the LV remodelling. However, our findings do not confirm any differences in oxidative/antioxidant markers between the groups, according to both the LV parameters and the LV function.

Šaric et al. investigated markers of protein oxidation, including advanced oxidation protein products (AOPP) and P-SH concentrations in serum. They found that these markers were higher in patients with CHF (*n* = 81) compared to healthy subjects (*n* = 68) (AOPP averages were 98.52 (26.52) µmol/mg and 49.83 (22.34) µmol/mg, respectively, *p* < 0.001) [30]; P-SH–295.24 (94.87) µmol/L and 215.24 (94.01) µmol/L). Markers of lipid peroxidation, reactive thiobarbituric acid substances (TBARS), were also found to be higher in the CHF group compared to healthy controls (TBARS averages were 19.47 (7.37) µmol/L and 17.06 (9.56) µmol/L, respectively, *p* < 0.005). Consequently, the concentration of TBARS was higher in cLVH compared to NG (TBARS averages were 23.35 (8.24) µmol/L and 16.91 (8.70) µmol/L, respectively, *p* < 0.05) [32x]. These findings are consistent with the results of Zorica et al., who found that MDA was higher in patients with hemodialysed cLVH [20]. Catalase activity did not differ between CHF and controls in this study but was higher in eLVH compared to cLVH (catalase averages were 53.03 (20.40) U/L and 74.05 (31.29) U/L, respectively, *p* < 0.05) [30]. GSH activity was significantly lower in the NYHA III/IV class (with *n* = 28 and *n* = 10, respectively) compared to patients with asymptomatic CHF (NYHA I with *n* = 11 and NYHA II with *n* = 71), as well as to healthy people (*n* = 60). SOD was found to increase in all CHF groups compared to healthy in another study [29]. Once again, in our CHF group, both protein and lipid oxidation products, as well as catalase activity and TAC, did not differ between the groups according to LV geometry or function.

In summary, the results of oxidative stress/antioxidant markers in CVD are obtained from different groups of patients using different methodologies (including not standardised methods for TBARS, AOPP, P-SH), as well as the small number of cases examined. Despite these differences, most studies have found increased protein and lipid oxidation in CVD patients and deterioration of the activity of enzymes of the antioxidant system. However, it is difficult to compare oxidative stress/antioxidant markers in CHF patients based on changes in the geometry of the LV and function, as we only found a few works related to it, and only one of them analysed CHF patients. Our work revealed no differences in serum concentrations of both oxidant (NT-Tyr, PC, MDA) and antioxidant (TAC and catalase) concentrations in CHF patients’ groups according to LV function and geometry. It should be mentioned that in our work, we used standardised methods and a homogeneous group of patients with CHF. Our results cannot confirm both our hypothesis that in HFpEF, there should be more oxidised proteins and lipids in the blood than in HFrEF, and that the concentration of oxidative stress markers in the serum could differ in the groups of patients according to the geometry of the LV.

### 4.2. Relationship between Oxidative Stress and Lipid Metabolism Markers

Investigation of the relationship between oxidative stress and markers of lipid metabolism showed interesting findings. Significant positive associations were found between MDA concentration and total cholesterol (*r_s_* = 0.337, *p* = 0.008), LDL cholesterol (*r_s_* = 0.295, *p* = 0.022), and non-HDL cholesterol (*r_s_* = 0.301, *p* = 0.019) levels, demonstrating that higher cholesterol concentration are associated with higher lipid peroxidation levels.

In our study group, NT-Tyr concentration negatively correlated with HDL cholesterol concentration (*r_s_* = −0.285, *p* = 0.027) and positively correlated with oxHDL (*r_s_* = 0.278, *p* = 0.0314). This finding could suggest that HDL is important for inhibiting protein oxidation.

Previous works in rats and apoB-100 transgenic mice have shown that hypercholesterolemia increases nitro-oxidative stress leading to myocardial function deterioration [31]. The same authors found in their other works that in patients with CAD (*n* = 36), the concentration of NT-Tyr in the blood was positively correlated with triglyceride (TAG) *r* = 0.47; *p* < 0.05), total (*r* = 0.58; *p* < 0.01), and LDL cholesterol levels (*r* = 0.45, *p* < 0.05), and negatively with HDL cholesterol (r = −0.46; *p* < 0.05) [32]. It is partially in agreement with our study, where in CHF patients, we found that NT-Tyr was negatively correlated with HDL and positively correlated with oxHDL cholesterol concentrations. The results, which are in agreement with those of other studies, could lead to the conclusion that the concentration of lipids in the blood may be related to the oxidation of proteins there. More detailed studies are needed to clarify this relationship.

### 4.3. Correlations between Lipid Metabolism Markers and LV Parameters

Some correlations were found in the CHF patients’ groups between lipid metabolism and LV markers. LVEDD, LVEDV, and LVESV were negatively correlated with HDL cholesterol concentration (*r* = −0.256, *p* < 0.05, *r* = −0.935, *p* < 0.0001, *r* = –0.906, *p* < 0.00001, respectively). IVST and LVWT were positively correlated with TAG concentration (*r* = 0.346, *p* < 0.01; *r* = 0.329, *p* < 0.05, respectively).

There are only a few publications on the relationship between lipid metabolism and LV parameters. Bencsikwith et al. did not observe any significant correlation between serum cholesterol and TAG levels with LVEF in the coronary artery disease group (*n* = 36) [32]. Wang et al. [33] described a positive correlation of LVEF with serum HDL cholesterol (*r* = 0.49, *p* < 0.0001) in patients with angina (*n* = 114). Dabas et al. reported no correlation between LV parameters and lipids in type one diabetes (*n* = 30) [34]. Al-Daydamony et al. found a significant positive correlation between TAG levels and LVMI in patients with metabolic syndrome [35].

The pathophysiological mechanism of these results cannot be explained simply. Hypercholesterolemia could adversely influence LV systolic function through its atherogenic effect on the restriction of coronary circulation. HDL cholesterol has been shown to reduce the risk of CAD at any concentration of LDL cholesterol [36] and remains a risk factor even for people with low serum total cholesterol and TAG concentrations [37]. This could explain why HDL cholesterol is more significant for LVEDD, LVEDV and LVESV than LDL in our study. Additionally, some studies revealed that lipids can accumulate directly around myocytes and act through other pathways. VLDL were presented to promote aldosterone overproduction that may interfere with LV remodelling [38]. Increased plasma aldosterone concentration was shown to be associated with increased TAG and decreased HDL cholesterol concentration in people with metabolic syndrome [39]. We did not measure both VLDL and aldosterone levels, so we cannot evaluate the potential pathway activated by VLDL that leads to overproduction of aldosterone and the development of LVH [38]. It should be mentioned that some research has reported the possibility of aldosterone to trigger activation of nicotinamide riboside kinase (nRK1/2) and p38 mitogen-activated protein kinase (MAPK), which have been involved in the signal transduction pathway associated with cardiac hypertrophy [40]. VLDLs are known to be a major transporter of TAGs, making up about 85% of their weight.

In summary, it could be stated that LV geometry could be related to lipid metabolism in patients with CHF, but future investigations are needed to reveal the possible TAG and HDL action pathways in the myocardium.

### 4.4. Correlations between Oxidative/Antioxidative and LV Markers

PC was found to be significantly correlated with LVEDD and LVEDV (*p* < 0.001 and *p* < 0.05, respectively), as well as TAC negatively correlated with both cLVH and eLVH phenotypes (*p* < 0.05 in both cases) in hemodialysis patients (*n* = 104) [20]. A weak positive correlation (*r* = 0.329; *p*< 0.05) was found between the levels of the P-SH group and the LVEF (*n* = 73) [28]. A significant association was found between plasma reactive carbonyl derivative levels (RCD) and the degree of LV remodelling (LVEDV and LVESV) only in patients with symptomatic CHF (*r* = 0.469; *p* < 0.008 and *r* = 0.452; *p* < 0.011, respectively) in another study [29]. Zarica et al. discussed that associations between ROS and LV indices prove that protein damage takes place in structural changes of the myocardium. Oxidized proteins and advanced glycation end products (AGE) has been stated as structural homology, and thereby oxidized proteins may serve as ligands for AGE receptors (RAGE) [41]. RAGE signalling activating the TGF-β pathway [42] activates cardiac remodelling [43].

The most important protein groups have been presented according to their functions related to nitrative modifications in heart tissue cells [44]. They involve proteins of the Tricarboxylic acid cycle (23%), lipid metabolism (9%), apoptotic process (9%), muscle contraction (9%), ATP biosynthetic process (9%) and others. The nitration of tyrosine residues in proteins was shown to inhibit protein catalytic activity [9] and damage the energy metabolism pathway [45] resulting in damage to heart function [46]. It was revealed that one of the enzymes of the intracellular antioxidant system, catalase, protects mouse hearts against cardiac remodelling by suppressing the intracellular NF-ĸB signalling pathway associated with protein nitration [44]. Additionally, Zorica et al. summarised that ROS entering the cell triggers the NF-kB pathway, resulting in increased cyclooxygenase 2 synthesis leading to cardiovascular inflammation and remodelling [47].

MDA was found to be significantly correlated with LVEDD and LVEDV (*p* < 0.001 and *p* < 0.05, respectively) in hemodialysed patients [20]. MDA and 8-epi-PGF2α significantly correlated with LVEF (*r* = –0.476; *p*< 0.001 and *r*= –0.787; *p*< 0.001, respectively) in Radovanovic’s CHF study [28]. Both results show a clear relationship between the level of oxidative lipids and the severity of myocardial dysfunction [28]. However, no significant correlation was found between antioxidant enzyme activities and ventricular remodelling indices in the research by Radovanovic et al. [29] and in ours. Unfortunately, in our CHF patients’ group, we did not find any correlation between oxidative stress and LV geometry or function markers. The reason for the discrepancies obtained may be the too small number of cases in our study or the different groups of patients investigated.

### 4.5. Limitations of the Study

Our work has some limitations. Because the study was pilot, we chose to investigate a small number of cases. We measured only a few markers of oxidative damage of proteins (NT-Tyr, dityrosine, protein carbonyl), lipids (MDA, oxHDL), and antioxidative (TPA, catalase). Investigation of 8-epi-PGF2α, P-SH, RCD concentrations, and antioxidant enzymes glutathione peroxidase and superoxide dismutase activity should be useful for a more complete picture. Measurements of serum aldosterone and TNF-α concentration would also have been useful. We did not evaluate diastolic function (mitral inflow E- and A-wave velocities). However, this is the first study investigating possible relationships in the pathogenesis of LV remodelling and oxidative/antioxidant markers. Future studies with a large sample size and involving markers of endocrine (aldosterone)-inflammation (TNF-α) markers in CHF developed due to ischaemic heart disease are required to explore the relationship.

## 5. Conclusions

Our work revealed no difference in serum concentrations of oxidant markers (NT-Tyr, PC, MDA) and antioxidant markers (TAC and catalase) in CHF patients’ groups according to LV function and geometry. While the geometry of the LV could be related to lipid metabolism in CHF patients, future investigations are needed to reveal the pathways through which TAG and HDL affect cardiomyocytes. Additionally, we did not find any correlation between oxidative/antioxidant markers and LV markers in CHF patients. Our hypothesis that in HFpEF there should be more oxidised proteins and lipids in the blood than in HFrEF patients, and that the concentration of oxidative stress markers in serum could differ in the groups of patients according to the geometry of the LV cannot be confirmed.

## Figures and Tables

**Table 1 cells-12-00803-t001:** Sociodemographic and clinical characteristics of the study group.

Variable	Median (IQR) or *n* (%)	Left Ventricular Ejection Fraction		χ^2^	df	*p*-Value
		<40% (*n* = 27)	≥40% (*n* = 33)			
Women	26 (43.33%)	7 (26.9%)	19 (73.1%)	6.06	1	0.014
Men	34 (56.67%)	20 (58.8%)	14 (41.2%)			
Age (years)	67.5 (22.5)	70 (21.5)	65 (22.0)			0.603
BMI (kg/m^2^)	26.00 (4.65) ^1^	25.85 (3.90) ^2^	26.00 (5.85) ^2^			0.975
Systolic blood pressure (mmHg)	130 (20)	131 (20.5)	130 (14.0)			0.575
Diastolic blood pressure (mmHg)	80 (20)	80 (18)	80 (18)			0.433
Total cholesterol (mmol/L)	4.52 (1.56)	4.20 (1.32)	4.98 (1.63)			0.062
HDL cholesterol (mmol/L)	1.09 (0.44)	1.08 (0.62)	1.10 (0.27)			0.705
LDL-cholesterol (mmol/L)	2.89 (1.15)	2.69 (1.22)	3.04 (0.96)			0.158
Non-HDL cholesterol (mmol/L)	3.34 (1.39)	3.28 (1.20)	3.64 (1.30)			0.124
Triacylglycerols (mmol/L)	1.23 (0.86)	0.88 (0.51)	1.50 (0.72)			0.003
NYHA classification						
Class II	20 (33.33%)	3 (15.0%)	17 (85.0%)	17.374	2	<0.001
Class III	20 (33.33%)	8 (40.0%)	12 (60.0%)			
Class IV	20 (33.33%)	16 (80.0%)	4 (20%)			

Abbreviations: BMI, body mass index; HDL, high-density lipoproteins; LDL, low-density lipoproteins; IQR, interquartile range; NYHA, The New York Heart Association. ^1^ Variable has missing data (*n* = 36), ^2^ Variable has missing data (*n* = 18).

**Table 2 cells-12-00803-t002:** Comparison of medication usage between the groups based on left ventricular ejection fraction values.

Medication	*n* (%)	Left Ventricular Ejection Fraction		*χ^2^*, df = 1	*p*-Value
		<40% (*n* = 27)	≥40% (*n* = 33)		
ACE inhibitors	34 (56.7%)	16 (59.3%)	18 (54.5%)	0.134	0.714
β-blockers	29 (48.3%)	12 (44.4%)	17 (51.5%)	0.297	0.586
Diuretics	18 (30.0%)	7 (25.9%)	11 (33.3%)	0.388	0.533
Heparin	7 (11.7%)	4 (14.8%)	3 (9.1%)	0.472	0.690
Calcium channel blockers	5 (8.3%)	3 (11.1%)	2 (6.1%)	0.496	0.649
Digoxin	10 (16.7%)	6 (22.2%)	4 (12.1%)	1.091	0.322
Warfarin	13 (21.7%)	6 (22.2%)	7 (21.2%)	0.009	0.925

Abbreviations: ACE, Angiotensin-converting enzyme.

**Table 3 cells-12-00803-t003:** Comparison of oxidative/antioxidant stress markers between the groups based on values of the left ventricular ejection fraction.

Variable	Mean ± SD or median (IQR)	Left Ventricular Ejection Fraction		*p*-Value
		<40% (*n* = 27)	≥40% (*n* = 33)	
Nitrotyrosine (nM)	3.94 (2.62)	4.47 (2.58)	3.65 (2.47)	0.154
Dityrosine (RUF)	8.03 ± 1.76	7.98 ± 1.97	8.07 ± 1.60	0.849
Total plasma antioxidant capacity (U/mL)	0.59 (0.98)	0.68 (1.04)	0.54 (0.65)	0.316
Protein carbonyl (U/mL)	259.95 (117.00)	291.56 (159.31)	244.90 (92.31)	0.082
Catalase (U/mL)	137.68 ± 67.62	142.53 ± 68.60	133.71 ± 67.61	0.619
Malondialdehyde (µg/L)	116.24 ± 24.78	110.90 ± 25.21	120.60 ± 23.93	0.133
Oxidized HDL (pg/L)	3.06 (3.95)	3.44 (1.71)	2.89 (6.35)	0.417

Abbreviations: HDL, high-density lipoproteins; IQR, interquartile range; RUF, relative units of fluorescence; SD, standard deviation.

**Table 4 cells-12-00803-t004:** Comparison of oxidative/antioxidant stress markers and values of left ventricular ejection fraction between groups based on the level of malondialdehyde in blood serum.

Variable	Malondialdehyde		*p*-Value
	≤114.29 µg/L	>114.29 µg/L	
Nitrotyrosine (nM), median (IQR)	3.98 (3.30)	3.91 (2.18)	0.641
Dityrosine (RUF), mean ± SD	7.78 ± 1.74	8.28 ± 1.77	0.276
Total plasma antioxidant capacity (U/mL), median (IQR)	0.54 (0.62)	0.68 (1.08)	0.762
Protein carbonyl (U/mL), median (IQR)	246.54 (112.81)	279.10 (118.25)	0.492
Catalase (U/mL), mean ± SD	138.13 ± 61.78	137.23 ± 74.06	0.9595
Oxidized HDL (pg/L), median (IQR)	2.944 (3.23)	3.196 (4.28)	0.994
Left ventricular ejection fraction (%), median (IQR)	36.5 (21.25)	40.0 (15.75)	0.099

Abbreviations: HDL, high-density lipoproteins; IQR, interquartile range; RUF, Relative Units of Fluorescence; SD, standard deviation.

**Table 5 cells-12-00803-t005:** Comparison of oxidative/antioxidant stress markers and left ventricular ejection fraction values between groups according to the protein carbonyl level in the blood serum.

Variable	Protein carbonyl		*p*-Value
	≤259.95 U/mL	>259.95 U/mL	
Nitrotyrosine (nM), median (IQR)	3.16 (2.09)	4.46 (2.12)	0.008
Dityrosine (RUF), mean ± SD	7.69 ± 1.55	8.37 ± 1.91	0.132
Total plasma antioxidant capacity (U/mL), median (IQR)	0.66 (0.55)	0.38 (1.19)	0.636
Malondialdehyde (µg/L), mean ± SD	114.33 ± 22.23	118.14 ± 27.35	0.556
Catalase (U/mL), mean ± SD	146.76 ± 61.07	128.60 ± 73.50	0.3022
Oxidized HDL (pg/L), median (IQR)	2.82 (2.08)	3.37 (5.27)	0.390
Left ventricular ejection fraction (%), median (IQR)	40.0 (18.75)	36.5 (20.00)	0.252

Abbreviations: HDL, high-density lipoproteins; IQR, interquartile range; RUF, relative units of fluorescence; SD, standard deviation.

**Table 6 cells-12-00803-t006:** Comparison of oxidative/antioxidative stress markers and left ventricular ejection fraction values between the groups based on the oxidised HDL level in blood serum.

Variable	Oxidized HDL		*p*-Value
	≤3.06 pg/L	>3.06 pg/L	
Nitrotyrosine (nM), median (IQR)	3.42 (1.38)	4.51 (2.45)	0.004
Dityrosine (RUF), mean ± SD	8.25 ± 1.96	7.80 ± 1.53	0.3255
Total plasma antioxidant capacity (U/mL), median (IQR)	0.59 (0.965)	0.61 (0.810)	0.728
Malondialdehyde (µg/L), median (IQR)	113.32 (22.52)	116.83 (35.37)	0.741
Catalase (U/mL), mean ± SD	153.50 ± 68.13	121.87 ± 64.37	0.06965
Malondialdehyde (µg/L), mean ± SD	114.81 ± 20.91	117.66 ± 28.42	0.6594
Protein carbonyl (U/mL), median (IQR)	247.57 (112.22)	270.66 (175.72)	0.2088
Left ventricular ejection fraction (%), median (IQR)	40.0 (14.00)	35.0 (22.00)	0.4709

Abbreviations: HDL, high-density lipoproteins; IQR, interquartile range; RUF, relative units of fluorescence; SD, standard deviation.

**Table 7 cells-12-00803-t007:** Sociodemographic and clinical characteristics of the study group according to the geometry of the left ventricle.

Variable	NG *n* = 7	CR *n* = 14	cLVH *n* = 16	eLVH *n* = 23	χ^2^	df	*p*-Value
Women	2 (28.6%)	5 (35.7%)	12 (75.0%)	7 (30.4%)	9.04	3	0.030 *
Men	5 (71.4%)	9 (64.3%)	4 (25.0%)	16 (69.6%)			
Age (years)	51.0 (18.5)	74.5 (9.75)	75.5 (15.0)	58.0 (14.5)			0.003 **
BMI (kg/m^2^)	23.50 (2.75) ^1^	26.00 (4.05) ^2^	26.25 (5.83) ^2^	26.10 (4.62) ^3^			0.847
Systolic blood pressure (mmHg)	136 (10.0)	130 (5.00)	135.5 (14.75)	122 (20.0)			0.430
Diastolic blood pressure (mmHg)	80 (8.0)	81 (10.0)	80 (16.0)	80 (20.0)			0.264
Total cholesterol (mmol/l)	4.97 (1.18)	4.20 (1.83)	4.93 (1.42)	4.27 (1.52)			0.259
HDL-cholesterol (mmol/l)	1.58 (0.71)	1.04 (0.44)	1.25 (0.55)	1.06 (0.24)			0.035 ***
LDL-cholesterol (mmol/l)	3.04 (0.67)	2.43 (1.14)	3.05 (1.17)	2.87 (1.17)			0.190
Non-HDL-cholesterol (mmol/l)	3.39 (0.79)	3.22 (1.38)	3.38 (1.31)	3.33 (1.60)			0.639
Triacylglycerols (mmol/l)	0.73 (0.13)	1.25 (0.71)	1.50 (0.65)	1.17 (0.85)			0.084
Ischemic heart disease							
Yes	1 (14.3%)	8 (57.1%)	5 (31.3%)	11 (47.8%)	4.61	3	0.210
No	6 (85.7%)	6 (42.9%)	11(68.7%)	12 (52.2%)			
NYHA classification							
Class II	2 (28.6%)	7 (50.0%)	5 (31.2%)	6 (26.1%)	3.63	6	0.759
Class III	2 (28.6%)	3 (21.4%)	7 (43.8 %)	8 (34.8%)			
Class IV	3 (42.8%)	4 (28.6%)	4 (25.0%)	9 (39.1%)			

* Post hoc analysis for Pearson’s Chi-squared test: non-significant. ** Dunn’s test with Bonferroni correction: NG vs. cLVH, *p* = 0.013; eLVH vs. cLVH, *p* = 0.022. *** Dunn’s test with Bonferroni correction: non-significant. Abbreviations: BMI, body mass index; HDL, high-density lipoproteins; LDL, low-density lipoproteins; IQR, interquartile range; NYHA, New York Heart Association; NG, normal left ventricle geometry; CR, concentric remodelling; cLVH, concentric left ventricular hypertrophy; eLVF, eccentric left ventricular hypertrophy. ^1^ Variable has missing data (*n* = 4), ^2^ Variable has missing data (*n* = 10), ^3^ Variable has missing data (*n* = 12).

**Table 8 cells-12-00803-t008:** Comparison of medication use between groups based on the geometry of the left ventricle.

Medication	NG *n* = 7	CR *n* = 14	cLVH *n* = 16	eLVH *n* = 23	χ^2^, df = 3	*p*-Value
ACE inhibitors	4 (57.1%)	10 (71.4%)	9 (56.3%)	11 (47.8%)	1.976	0.565
β-blockers	3 (42.9%)	5 (35.7%)	11 (68.8%)	10 (43.5%)	3.865	0.277
Diuretics	2 (28.6%)	3 (21.4%)	5 (31.3%)	8 (34.8%)	0.759	0.897
Heparin	1 (14.3%)	2 (14.3%)	2 (12.5%)	2 (8.7%)	0.348	0.946
Calcium channel blockers	0 (0%)	2 (14.3%)	2 (12.5%)	1 (4.3%)	2.128	0.660
Digoxin	0 (0%)	2 (14.3%)	2 (12.5%)	6 (26.1%)	3.127	0.503
Warfarin	0 (0%)	1 (7.1%)	6 (37.5%)	6 (26.1%)	6.304	0.111

Abbreviations: ACE, angiotensin converting enzyme; CR, concentric remodelling; cLVH, concentric left ventricular hypertrophy; eLVF, eccentric left ventricular hypertrophy.

**Table 9 cells-12-00803-t009:** Comparison of oxidative/antioxidative stress markers between the groups based on the left ventricular geometry.

Variable	NG *n* = 7	CR *n* = 14	cLVH *n* = 16	eLVH *n* = 23	*p*-Value
Nitrotyrosine (nM), median (IQR)	4.01 (1.33)	4.57 (2.67)	2.70 (1.81)	4.47 (2.40)	0.086
Dityrosine (RUF), median (IQR)	8.69 (2.86)	7.93 (1.79)	8.29 (1.41)	7.75 (3.01)	0.996
Total plasma antioxidant capacity (U/mL), median (IQR)	0.36 (0.35)	0.89 (0.99)	0.65 (0.47)	0.37 (1.08)	0.361
Protein carbonyl (U/mL), median (IQR)	184.03 (117.73)	259.54 (74.08)	242.66 (73.66)	292.00 (167.40)	0.159
Catalase (U/mL), median (IQR)	118.20 (29.56)	132.97 (70.19)	147.75 (57.26)	132.97 (75.36)	0.743
Malondialdehyde (µg/L), median (IQR)	119.28 (14.89)	117.42 (38.99)	112.37 (23.73)	114.21 (37.00)	0.813
Oxidized HDL (pg/L), median (IQR)	4.91 (4.27)	2.85 (3.92)	2.65 (4.90)	3.00 (1.89)	0.345

Abbreviations: HDL, high-density lipoproteins; IQR, interquartile range; RUF, relative units of fluorescence; SD, standard deviation; CR, concentric remodelling; cLVH, concentric left ventricular hypertrophy; eLVF, eccentric left ventricular hypertrophy.

**Table 10 cells-12-00803-t010:** Correlations between oxidative/antioxidative stress markers and values of left ventricular ejection fraction.

Variable	Ejection Fraction (%)			
	Spearman’s *r*	*p*-Value	Spearman’s *r* (Gender-Adjusted)	*p*-Value
Nitrotyrosine (nM)	−0.185	0.158	−0.135	0.309
Dityrosine (RUF)	−0.055	0.674	−0.068	0.606
Total plasma antioxidant capacity (U/mL)	−0.085	0.518	−0.041	0.760
Protein carbonyl (U/mL)	−0.257	0.047	−0.240	0.067
Catalase (U/mL)	−0.071	0.588	−0.120	0.365
Malondialdehyde (µg/L)	0.176	0.180	0.103	0.436
Oxidized HDL (pg/L)	−0.100	0.446	−0.096	0.470

Abbreviations: RUF, Relative units of fluorescence; HDL, high-density lipoproteins.

**Table 11 cells-12-00803-t011:** Correlation matrix of oxidative/antioxidative stress markers.

Variable	NT-Tyr	di-Tyr	TAC	PC	CAT	MDA	oxHDL
NT-Tyr	1.000						
di-Tyr	0.066	1.000					
TAC	0.124	−0.190	1.000				
PC	0.482 **	0.199	0.005	1.000			
CAT	−0.192	−0.170	−0.165	−0.100	1.000		
MDA	−0.212	0.109	0.031	0.152	0.162	1.000	
oxHDL	0.278 *	−0.203	0.003	0.137	−0.297	−0.063	1.000

* *p* < 0.05. ** *p*< 0.0001. Abbreviations: NT-Tyr, nitrotyrosine; di-Tyr, dityrosine; TAC, total plasma antioxidant capacity; PC, protein carbonyl; CAT, catalase; MDA, malondialdehyde, oxHDL, oxidized HDL.

**Table 12 cells-12-00803-t012:** Correlations between lipid metabolism biomarkers and clinical data or oxidative/antioxidative stress markers.

Variable	Age	SBP	DBP	NT-Tyr	di-Tyr	TAC	CAT	MDA	oxHDL
TC	0.125	0.255 *	0.231	−0.199	−0.011	0.021	0.096	0.337 **	−0.038
HDL-C	0.315 *	0.246	0.167	−0.285 *	−0.111	−0.076	0.041	0.232	0.045
LDL-C	−0.020	0.145	0.150	−0.161	0.043	−0.025	0.023	0.295 *	−0.037
Non-HDL-C	0.052	0.184	0.206	−0.123	0.040	0.040	0.079	0.301 *	−0.056
TAG	0.055	0.025	0.133	−0.083	0.047	0.218	0.135	0.153	−0.205

** p* < 0.05. ** *p* < 0.01. Abbreviations: SBP, systolic blood pressure; DBP, diastolic blood pressure; LVEF, Left ventricular ejection fraction; NT-Tyr, nitrotyrosine; di-Tyr, dityrosine; TAC, total plasma antioxidant capacity; PC, protein carbonyl; CAT, catalase; MDA, malondialdehyde; oxHDL, oxidized HDL, TC, total cholesterol; HDL-C, high-density lipoprotein cholesterol; non-HDL-C, non-high-density lipoprotein cholesterol; TAG, triacylglycerols.

**Table 13 cells-12-00803-t013:** Correlations between oxidative/antioxidative stress markers and echocardiographic characteristics.

Variable	LVEDD	IVST	PWT	LVWT	RWT	LVM	LVMI	LVEDV	LVESV
NT-Tyr	0.229	–0.112	0.133	–0.005	–0.145	0.188	0.179	0.289	0.280
di-Tyr	0.032	–0.052	–0.022	–0.032	0.019	–0.040	0.016	0.024	0.085
TAC	0.038	0.059	0.087	0.061	0.114	0.061	0.067	0.287	0.341
PC	0.198	0.030	0.005	0.014	–0.135	0.188	0.234	0.190	0.159
CAT	–0.051	0.218	0.049	0.155	0.063	–0.005	–0.073	0.246	0.249
MDA	–0.172	–0.056	–0.083	–0.075	0.069	–0.198	–0.141	–0.368	–0.385
oxHDL	–0.148	–0.065	0.014	–0.022	–0.001	–0.054	–0.024	0.085	0.082

Abbreviations: NT-Tyr, nitrotyrosine; di-Tyr, dityrosine; TAC, total plasma antioxidant capacity; PC, protein carbonyl; CAT, catalase; MDA, malondialdehyde; oxHDL, oxidized HDL; LVEDD, left ventricular end-diastolic dimension; IVST, interventricular septum thickness; PWT, posterior wall thickness; LVWT, left ventricular wall thickness; RWT, relative wall thickness; LVM, left ventricular mass; LVMI, left ventricular mass index; LVEDV, left ventricular end-diastolic volume; LVESV, left ventricular end-systolic volume.

**Table 14 cells-12-00803-t014:** Correlations between sociodemographic, clinical data or lipid metabolism biomarkers and echocardiographic characteristics.

Variable	LVEDD	IVST	PWT	LVWT	RWT	LVM	LVMI	LVEDV	LVESV
Age	–0.335 **	0.448 ***	0.383 **	0.434 ***	0.461 ***	0.004	–0.019	–0.481	–0.460
SBP	–0.141	0.205	0.092	0.162	0.199	–0.103	–0.167	–0.492	–0.485
DBP	–0.007	0.099	–0.065	0.014	–0.002	–0.032	–0.245	0.068	0.081
TC	–0.186	0.227	0.056	0.162	0.152	–0.112	–0.106	–0.150	–0.088
HDL-C	–0.256 *	0.202	–0.020	0.119	0.151	–0.197	–0.166	–0.935 ****	–0.906 ****
LDL-C	–0.130	0.065	–0.013	0.034	0.075	–0.131	–0.115	0.091	0.159
Non-HDL-C	–0.107	0.187	0.071	0.145	0.095	–0.027	–0.046	0.146	0.209
TAG	–0.121	0.346 **	0.244	0.329 *	0.208	0.044	–0.010	0.364	0.368

* *p* < 0.05. ** *p* < 0.01 *** *p* < 0.001 **** *p* < 0.0001. Abbreviations: SBP, systolic blood pressure; DBP, diastolic blood pressure; TC, total cholesterol; HDL-C, high-density lipoprotein cholesterol; non-HDL-C, non-high-density lipoprotein cholesterol; TAG, triacylglycerols; LVEDD, left ventricular end-diastolic dimension; IVST, interventricular septum thickness; PWT, posterior wall thickness; LVWT, left ventricular wall thickness; RWT, relative wall thickness; LVM, left ventricular mass; LVMI, left ventricular mass index; LVEDV, left ventricular end-diastolic volume; LVESV, left ventricular end-systolic volume.

## Data Availability

The study did not report any data.

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
