# Peer review of "Relationship between Oxidative Stress and Left Ventricle Markers in Patients with Chronic Heart Failure"

_cells, 2023, doi:10.3390/cells12050803_

Round 1

Reviewer 1 Report

As a reviewer, I would like to congratulate the authors for putting together their interesting observations. being said I have couple of doubts/corrections in the manuscript. 

1. in page 3, line 100, do authors mean GFR (glomerular filtration rate)? if not please explain the fill form.

2. in page 4, line 168, expand ACE in full form. I am aware authors described it under the table abbreviations, still authors need to include once in the main text.

3. Although authors highlighted the lack of control group, I was thinking what could be the control group data across the manuscript? If authors can present some data in this direction, manuscript will be more interesting.  

Author Response

Thanks to the Reviewer for reviewing the article and helpful comments. We have corrected the article in the light of the comments. It improved the quality of the article. The following corrections have been made.

1.in page 3, line 100, do authors mean GFR (glomerular filtration rate)? if not please explain the fill form.

GFR authors mean glomerular filtration rate. The full form in page 3, line 100 has been added.

  1. in page 4, line 168, expand ACE in full form. I am aware authors described it under the table abbreviations, still authors need to include once in the main text.

In page 4, line 168, expand ACE in full form has been added. Under the table abbreviations it has been explained too.

  1. Although authors highlighted the lack of control group, I was thinking what could be the control group data across the manuscript? If authors can present some data in this direction, manuscript will be more interesting.  

We thought that group of healthy person could be the control group, but, we do not have  it. On the other hand, the aim of work was to compare CHF groups, so the healthy group seems not necessary for it. According to comment we removed this from the text. Please see the page 13 line 517.

Reviewer 2 Report

The subject of the article is extremely interesting, with implications regarding both the pathophysiology of heart failure, as well as practical applicability aiming  new methods of prognostic evaluation of the patient with heart failure but also finding new therapeutic targets.

From a physiopathological point of view, heart failure offers an extremely vast field for exploration and from this point of view the article is welcome due to the correlations it supports between the biochemical, echocardiographic and clinical parameters.

This can be useful for clinical practice, helping the complex evaluation of patients with heart failure.

Some observations regarding the formulation of the phrases were inserted into the text.

Some references are old and very old and also not very closed to the subject (41, 42)

I suggest another term instead of ”readings” . A more ”medical” term

Author Response

Thanks to the Reviewer for reviewing the article and helpful comments. We have corrected the article in the light of the comments. It improved the quality of the article. The following corrections have been made.

Some references are old and very old and also not very closed to the subject (41, 42)

According to comment, this part has been removed from the text. Please see page 12, lines 469-477.

I suggest another term instead of ”readings” . A more ”medical” term

Tearm „readings“ has been changed to „markers“.

Round 2

Reviewer 1 Report

I would like to thank the editor for sending the manuscript for revision. When it comes to manuscript authors made necessary changes I had advised before. Although control groups are missing I am convinced with the authors explanation. 

I would recommend the manuscript for publication as it present a interesting prospective.